

# A molecular classification of human mesenchymal stromal cells

Florian Rohart[1,2], Elizabeth A. Mason[1], Nicholas Matigian[1,2],
Rowland Mosbergen[1,3], Othmar Korn[1], Tyrone Chen[1,3], Suzanne Butcher[1,3],
Jatin Patel[4], Kerry Atkinson[4], Kiarash Khosrotehrani[4,5], Nicholas M. Fisk[4,5],
Kim-Anh Lê Cao[2] and Christine A. Wells[1,3]

[1] Australian Institute for Bioengineering and Nanotechnology, University of Queensland, Brisbane, Queensland, Australia
[2] The University of Queensland Diamantina Institute, Translational Research Institute, University of Queensland, Brisbane, Queensland, Australia
[3] Department of Anatomy and Neuroscience, Faculty of Medicine, University of Melbourne, Melbourne, Victoria, Australia
[4] The University of Queensland Centre for Clinical Research, University of Queensland, Brisbane, Queensland, Australia
[5] Centre for Advanced Prenatal Care, Royal Brisbane & Women's Hospital, Brisbane, Queensland, Australia

Corresponding author
Christine A. Wells,
wells.c@unimelb.edu.au

## ABSTRACT

Mesenchymal stromal cells (MSC) are widely used for the study of mesenchymal tissue repair, and increasingly adopted for cell therapy, despite the lack of consensus on the identity of these cells. In part this is due to the lack of specificity of MSC markers. Distinguishing MSC from other stromal cells such as fibroblasts is particularly difficult using standard analysis of surface proteins, and there is an urgent need for improved classification approaches. Transcriptome profiling is commonly used to describe and compare different cell types; however, efforts to identify specific markers of rare cellular subsets may be confounded by the small sample sizes of most studies. Consequently, it is difficult to derive reproducible, and therefore useful markers. We addressed the question of MSC classification with a large integrative analysis of many public MSC datasets. We derived a sparse classifier (The Rohart MSC test) that accurately distinguished MSC from non-MSC samples with >97% accuracy on an internal training set of 635 samples from 41 studies derived on 10 different microarray platforms. The classifier was validated on an external test set of 1,291 samples from 65 studies derived on 15 different platforms, with >95% accuracy. The genes that contribute to the MSC classifier formed a protein-interaction network that included known MSC markers. Further evidence of the relevance of this new MSC panel came from the high number of Mendelian disorders associated with mutations in more than 65% of the network. These result in mesenchymal defects, particularly impacting on skeletal growth and function. The Rohart MSC test is a simple *in silico* test that accurately discriminates MSC from fibroblasts, other adult stem/progenitor cell types or differentiated stromal cells. It has been implemented in the www.stemformatics.org resource, to assist researchers wishing to benchmark their own MSC datasets or data from the public domain. The code is available from the CRAN repository and all data used to generate the MSC test is available to download via the Gene Expression Omnibus or the Stemformatics resource.

## INTRODUCTION

Adult tissues maintain the capacity to be replenished as part of the normal processes of homeostasis and repair. The adult stem cell hypothesis proposes that multipotent cells resident in tissues are the source of this cellular renewal, and expand in response to tissue injury. MSC were first isolated from bone marrow, where these occupy an important stem cell niche required for reconstitution of bone and the stromal compartments of marrow, and also play a supportive role in haematopoiesis (*Friedenstein, Piatetzky-Shapiro & Petrakova, 1966*; *Pittenger et al., 1999*). Subsequently, adult stromal progenitors have been isolated and cultured from most organs including placenta, heart, adipose tissue and kidneys although the identity of these cells remains controversial (reviewed by *Phinney, 2012*; *Bianco et al., 2013*). Specifically, the question of how similar cells isolated outside the bone marrow niche are, and whether these could be considered *bona fide* MSC, or indeed, challengingly, whether MSC isolated from different tissues share any phenotypic or molecular characteristics at all (*Bianco et al., 2013*). In this light, various cells described as MSC (whether by name or attribution) have been reported as having quite different self-renewal capacity, immunomodulatory properties or propensity to differentiate *in vivo* (*Reinisch et al., 2014*). It has been variously argued that MSC isolated from most stromal tissues are derived from perivascular progenitors (*Crisan et al., 2008*), or recruited from the bone marrow to distal tissue sites (*Lee et al., 2010*), or that resident stromal progenitors from different tissues must have tissue-restricted phenotypes. The most stringent criterion for MSC are *in-vivo*, bone-forming capacity; however, this functional standard is rarely addressed in the majority of MSC studies reported in the literature to date (see for example *Reinisch et al., 2014*; *Sworder et al., 2015*).

Several groups have attempted to address the demand for improved molecular markers, for example using global proteomics methods (*Li et al., 2009*), epigenetic markers (*De Almeida et al., 2016*), transcriptome analysis of cells capable of regenerating the bone marrow niche (*Charbord et al., 2015*), or comparison of desirable properties such as capacity to form bone (*Sworder et al., 2015*) and indeed the studies reporting global 'omic' analysis of MSC number in the hundreds. Each of these studies identifies a different set of potential markers, but there is little consensus among them. Most human studies have been conducted on very small numbers of donors, so it is difficult to dissect donor-donor heterogeneity from source heterogeneity. Nevertheless, variation between MSC lines is a major contributor to differences in MSC growth and differentiation capacity, and clonal variation is evident even when derived from the same donor bone marrow (*Samsonraj et al., 2015*; *Sworder et al., 2015*). MSC heterogeneity is further compounded by growth conditions, including the density of culture, the inclusion of serum, or the substrate on which they are grown (*Liu et al., 2015*). Consequently there is little agreement in the literature on definitive molecular or cellular phenotypes of human cultured MSC, whether from bone marrow or other sources.

There is little consensus on whether MSC from differing tissue sources share common functional attributes. The lack of definitive markers for human MSC is a major barrier to understanding genuine similarities, or resolving differences between various cell sources or

subsets. Even if acknowledging that there should be functional differences between MSC isolated from different tissues, or donor groups, it is not clear whether there should be any over-arching commonalities that might indicate shared homeostatic roles or ontogenies. The field requires improved methods for benchmarking MSC cultures, including molecular methods that lack the ambiguity of current antibody-based methods. Here we describe a sophisticated integrative transcriptome analysis of public MSC datasets, and provide a highly accurate *in silico* tool for straightforward assessment of the identity of an MSC culture.

## MATERIAL AND METHODS

### Design of test and training datasets

A careful screening of all the datasets collated in www.stemformatics.org (*Wells et al., 2012*), GEO (*Barrett et al., 2011*) and ArrayExpress (*Parkinson et al., 2011*) at the time of this analysis identified 120 possible MSC microarray datasets. These were evaluated for the availability of the primary (unprocessed) data; unambiguous replication (biological not technical); the quality control metrics of RNA quality ($5'-3'$ probe ratios); linear range (box-whisker plots of sample median, min and max absolute and normalized values); unambiguous sample descriptions; and sample clustering concordant with the original publication. 35/120 datasets failed these criteria and were excluded from the study.

As the range of phenotypes employed across the remaining 85 MSC microarray studies was broad (Table S2), we assigned to the training group only those MSC datasets that met at least the following criteria in common: Adherence, Cell surface markers CD105+, CD73+, CD45− and differentiation to at least two of the three MSC-definitive lineages (bone, cartilage or fat), and all training datasets included substantial phenotyping above these minimal criteria. These minimal common criteria were hard-coded into the Stemformatics annotation pipeline, we had a dedicated annotator responsible for the quality of these annotations and these were reviewed independently by two additional annotators. Sixteen MSC datasets met our 'gold standard' training set criteria for accompanying phenotype of MSCs, together with 27 datasets containing cells from non-mesenchymal or non-stromal sources, which we refer to as non-MSCs. In total, 41 datasets were included in the training set, with two datasets containing both MSCs and non-MSCs, with a total of 125 MSC samples and 510 non-MSC samples from 10 different microarray platforms (Table S3, accompanies the MSC clustering in Fig. 2). The remaining MSC datasets were assigned to the independent test set and were used only for evaluation of accuracy of the final signature.

Details on the samples, datasets and references of the experiments can be found in Tables S2, S3 and S5. Two large datasets—5003 (211 non-MSCs) and 6063 (45 MSCs), were subsampled prior to assigning to the training set to avoid unbalanced results. The samples left out were included in the test set (Table S5). It consisted of 65 experiments (1,291 samples, 213 MSCs and 499 non-MSC) profiled across 15 different platforms.

## Pre-processing of data

All data were processed using the R programming language v2.15.3 (*Venables, Smith & R Development Core Team, 2008*; *R Development Core Team R, 2011*). The pre-processing step involved a background correction performed with `affy 1.36.1` and the `affycoretools 1.30.0`, `gcrma 2.30.0`, `limma 3.14.4`, `lumi 2.10.0`, `simpleaffy 2.34.0` (*Gautier et al., 2004*; *Du, Kibbe & Lin, 2008*; *Carvalho & Irizarry, 2010*) packages for processing of microarray data depending on the platform.

Specifically, Affy GeneChips were background corrected using code:

```
data.bgonly <-
bg.adjust.gcrma(data,affinity.info=affinity_data,fast=FALSE)

## Extract GC-RMA bg-corrected expression values without re-running
additional bg-correction

data.bgexpr <- rma(data.bgonly, background=FALSE,normalize=FALSE)
```

where:
"data" is loaded raw CEL data
"affinity_data" is precomputed probe affinity produced by "`compute.affinities()`"

Affymetrix Gene ST arrays were RMA background corrected using `Affymetrix Power Tools v1.14.4.1` ("`apt_probeset_summarize`" tool). Exon probe expressions were summarised to the transcript level.

Illumina chips were background corrected using code:

```
lumiB(data, method = c('bgAdjust.affy'))
```

where:
"data" is non-normalized BeadStudio / GenomeStudio expression data returned by "`lumiR()`"

Agilent chips were background corrected using code:

```
dat <- backgroundCorrect(datraw, method=''normexp'',
normexp.method=''rma'')

datbg <- dat[ dat$ genes$ControlType==0, ]

bgave <- avereps(datbg, ID=datbg$ genes[,''ProbeName''])
```

where:
"datraw" is non-normalised Agilent data returned by "`read.maimages()`"

All data was subsequently log2 transformed and a YuGene transformation was applied (*Lê Cao et al., 2014*). YuGene is a rescaling method using the cumulative proportion that is applied per sample rather than per dataset or per series. This is highly advantageous as we performed 10-fold cross-validation that would otherwise require renormalization as datasets were added or removed.

In order to combine all the datasets described in Table S2, probes were mapped to Ensembl gene to provide a common set of identifiers. Mapping thresholds of 98% match were used to align microarray probes to Ensembl human v69 transcript model cDNA and ncRNA sequences obtained from Ensembl. Transcript IDs in resulting mapping were converted to Gene IDs using Ensembl BioMart v69 (*Zhang et al., 2011*). In the case of multi-mapping (several probes mapping to the same Ensembl gene ID), the probe with the highest average expression was chosen, on a per-dataset basis.

The combined training data set included the gene expression measurement of 41,185 genes mapped by at least one probe in one dataset. When a dataset had no probes mapping to a particular gene, the expression values of the gene were arbitrarily set to zero for all samples from that dataset. A pre-screening step was then performed to discard genes that were not present in at least half of the samples.

### Identification of the 16-gene signature and assignation of a test sample to the MSC or non-MSC class

The MSC signature was identified using a novel implementation of the sparse variant of Partial Least Square Discriminant Analysis (sPLS-DA) (*Barker & Rayens, 2003*) implemented for multiple microarray studies using the mixOmics package (*Lê Cao et al., 2009*; *Lê Cao, Boitard & Philippe, 2011*). Full details of the statistical model are provided in the Supplementary methods. The underlying code for the statistical test is available as BootsPLS in the CRAN repository, and we have also made available the d3 code for the interactive MSC graph implemented in Stemformatics via the BioJS framework at http://biojs.io/d/biojs-vis-rohart-msc-test.

### Network analysis

Twenty-six genes selected on component 1 equated to 20 proteins with a curated interaction in the NetworkAnalyst protein interaction database (which draws on the PPI database of the International Molecular Exchange (IMEx) consortium, accessed July 2015 (*Orchard et al., 2012*; *Xia, Benner & Hancock, 2014*)). These seed proteins were annotated to a shortest-path first-order network of 36 nodes (16 seeds) and 48 PPI edges. Twenty randomised sets of equivalent size were selected from the background (expressed) genes to demonstrate a lack of PPI structure by chance. Gene ontology analysis was assessed using hypergeometric mean against the Jan 2015 EBI UniProt GO library (*Huntley et al., 2015*) Disease annotations were undertaken using the OMIM (*Baxevanis, 2012*) and MGI (*Shaw, 2009*) databases. Subcellular location annotations were taken from UniProt (*EMBL, SIB Swiss Institute of Bioinformatics & Protein Information Resource (PIR), 2013*).

### Differential expression analysis

Individual MSC markers were assessed for differential analysis between MSC and non-MSC groups using a standard 2-tailed $t$-test, with a significance threshold of $10^{-6}$. For exploration of MSC subsets, a linear mixed model with dataset as random effect was fitted for each gene for which both the mean of bone marrow samples and other sites were higher than the median of all gene expression values. This retained 16,903 genes.

*P*-values were obtained by ANOVA and corrected for multiple testing with the Benjamini–Hochberg procedure (*Benjamini & Hochberg, 1995*).

## RESULTS

### Common MSC markers group MSC from bone marrow and other tissues

The International Society for Cellular Therapy (*Dominici et al., 2006*) has collated a large set of markers commonly used to immunophenotype MSC. These were used, in combination with more recently identified markers from the current literature (*Lv et al., 2014*), to assess whether a transcript-based approach might provide a useful molecular tool to identify MSC populations (Table S1). In order to compare data generated on different microarray platforms, we built a PLS-DA matrix using these markers and their corresponding expression in highly verified MSC samples. The resulting scatter plot (PLS-DA, Fig. 1A) demonstrated the capacity to distinguish between most MSC and non-MSC samples at a transcriptional level, and further showed that MSC isolated from different tissues do cluster together using these markers. Figure 1B shows the 16 of 32 commonly used MSC markers that were significantly differentially expressed between MSC and non-MSC groups ($P < 10^{-6}$), and these included CD73 (NTE5), CD105 (Endoglin), PDGFRB and VCAM1. The average expression of the remaining markers is provided in Fig. S1. Despite ISCT recommendations, most of the MSC publications reviewed herein used a small subset of these antibodies when phenotyping MSC, and CD73+, CD105+ and CD45− were the most consistent subset used (in combination with additional markers and phenotypic information; Table S2). When just these three markers were used to cluster all of the samples, 85% of MSC still grouped together (12/125 misclassified: Table 1, Fig. 1A), but almost 12% of non-MSC samples also clustered with this group. The overall accuracy increased to 92% when all 32 markers were used, but the rate of non-MSC misclassification remained high (7%, 35/510) and the majority of these (73.5%) were fibroblasts. It may be that these markers are less stably detected at a mRNA than protein level, however this high misclassification rate is also consistent with a large body of literature documenting the ambiguity of these markers, which are shared with stromal fibroblasts, endothelial progenitors and hematopoietic cells. The variable expression of all 32 markers (Fig. 1B, Fig. S1) is consistent with the reported variability of marker use in the wider MSC research community (reviewed by *Lv et al., 2014*; *Samsonraj et al., 2015*). Nevertheless, the capacity of these known markers to cluster MSC from different studies gave us confidence that a transcriptome approach was a useful and simplified alternate to antibody-based protocols, so we next took an unbiased approach to find a set of markers that could improve on the current classification paradigm. Our goal was to find an *in silico* marker set that reproducibly identified *bona fide* MSC samples regardless of platform or laboratory differences, and provide a molecular test that was simpler, and more accurate than current methods.

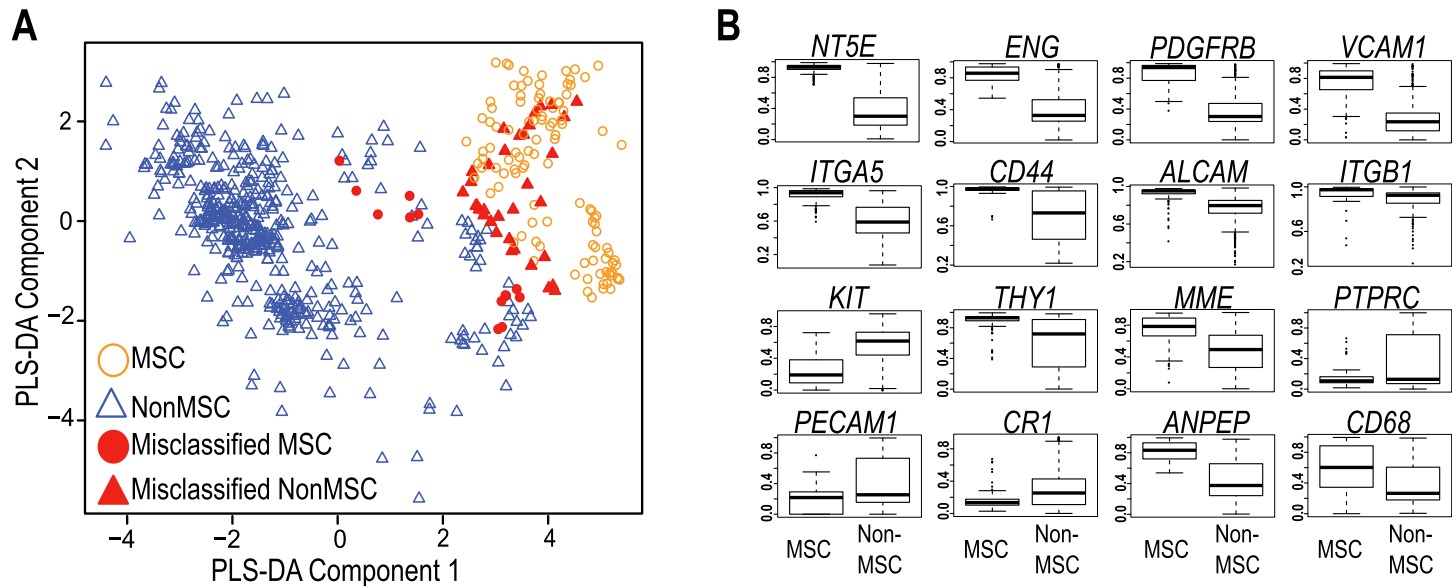

**Figure 1** **Evaluation of Common MSC markers as transcriptional classifiers.** (A) PLS-DA scatter plot of MSC (circles) and non-MSC cell types (triangles). Red symbols indicate cells which are incorrectly classified by the PLS-DA matrix. The matrix components consisted of 32 commonly used MSC markers. (B) Box and Whisker plots showing average expression of common MSC markers that are significantly differentially expressed ($t$-test, $P > 10^{-6}$) between MSC ($n = 125$) and non-MSC ($n = 510$) cell types. See also Fig. S1 and Table S1.

**Table 1** **MSC Signature improves the classification accuracy of MSC compared to a panel of 32 commonly used MSC markers.** Column 1 provides the comparison of the classification accuracy of the 635 training samples using (Column 2) the three markers used as the minimal immunophenotype of the MSC training samples. (Column 3) a panel of 32 commonly used immune-markers in the MSC literature; (Column 4) using the unrefined sPLS-DA output; or (Column 5) with our final signature of 16 genes. Performance of each gene group was assessed using 200 random subsamplings of the training set. The internal classification error rate was calculated from a PLS-DA with 2 components (known immune-markers), or was an output of our statistical model with genes selected in an unbiased manner (cf Fig. 1A).

|  | CD45, CD73, CD105 | 32 common MSC markers | sPLS-DA prior to stable gene selection | The 16-gene MSC signature |
|---|---|---|---|---|
| Overall accuracy (% of 635 samples) | 87.86 | 92.33 | 97.71 | 97.85 |
| MSC misclassified (% of 125 samples) | 14.40 | 11.10 | 3.04 | 4.31 |
| Non-MSC misclassified (% of 510 samples) | 11.60 | 6.82 | 2.11 | 1.61 |

## Derivation of an improved, simple and accurate in silico MSC classifier

A careful review of the public databases identified 120 potential MSC transcriptome studies, each comprising of a small number of donors. These were carefully curated for source, phenotypic information and growth conditions (see 'Methods' for details). From these efforts, a gold standard 'training set' was identified as meeting high confidence MSC phenotype including at least the minimal common set of CD73+, CD105+, CD45− and bilineage differentiation. The training set consisted of 125 MSC samples from 16 independently derived datasets derived predominantly from bone marrow, but also included studies from other adult, neonatal and fetal stromal sources. MSC were compared

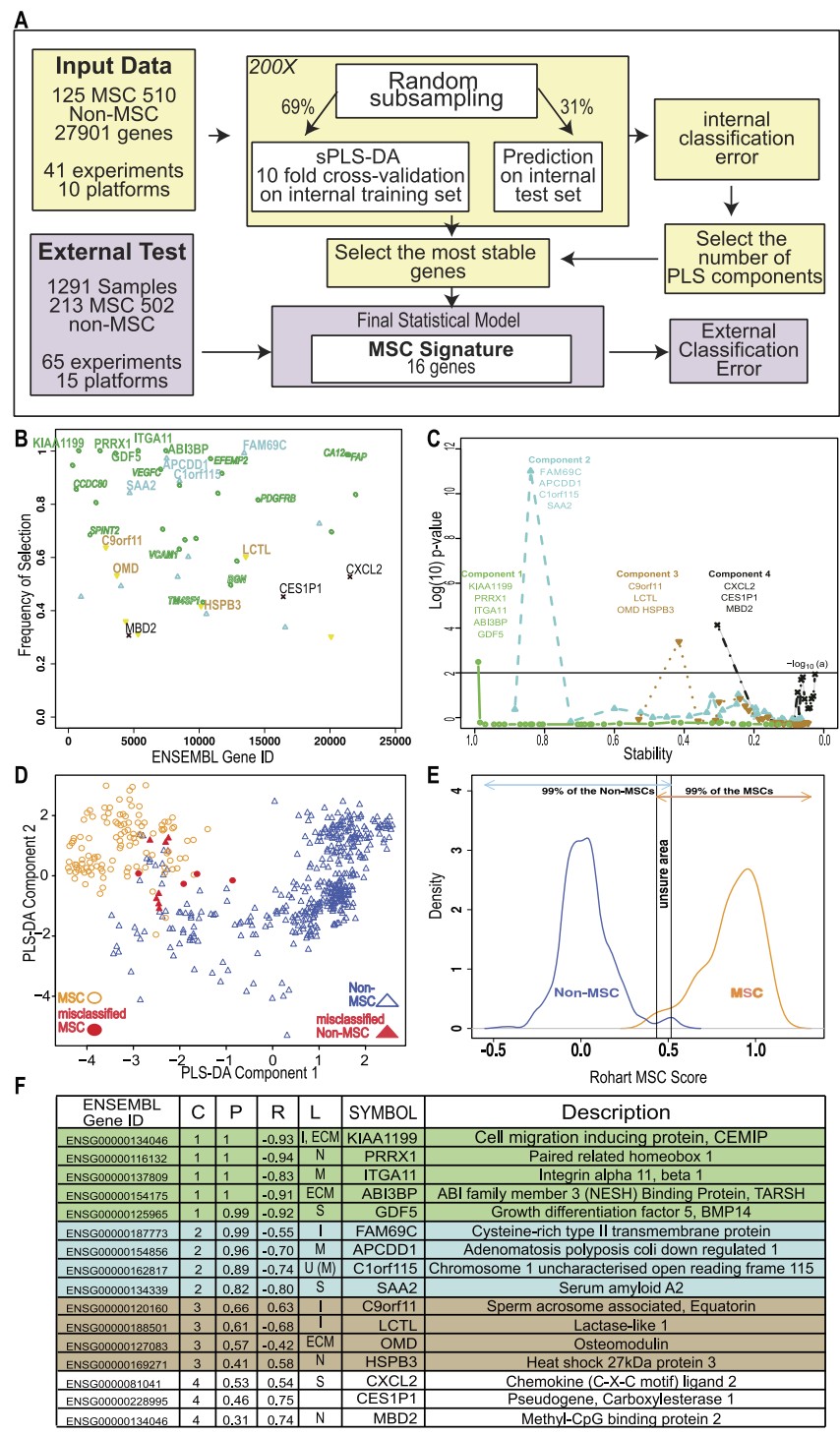

**Figure 2  An improved *in silico* MSC signature.** (A) Workflow summarizing the modified implementation of thesPLS-DA to integrate and evaluate cross-platform studies for derivation of a stable classifier; (B) Evaluation of the stability of each gene across four components, where frequency of selection over 200 subsamplings (*Y*-axis) is shown per gene (ENSEMBL ID, *X*-axis). Labels are provided for the 16 genes contributing to the signature across 4 components. Component 1 (green), (continued on next page...)
**Figure 2 (…continued)**
Component 2 (Blue), Component 3 (Brown), Component 4 (Black). Small text gene symbols indicate a selection of previously identified MSC markers that were excluded for poor stability. (C) Evaluation of the contribution of each gene to the informativeness of its component. Each dot is a gene set, ordered along the $x$-axis by decreasing stability (frequency of selection). The $y$-axis represents the $-\log_{10}$ ($P$-value) of a one tailed $t$-test indicating the improvement in classification accuracy across four components. (D) PLS-DA scatter plot showing sample clustering and classification accuracy of the training set (635 samples) in two components (Component 1 $X$-axis, Component 2 $Y$-axis). MSC samples are shown as circles, non-MSC as triangles, and misclassified samples are coloured red. (E) Identifying the scores that classify an MSC or non-MSC. Distribution of the Rohart MSC Score ($X$-axis) and the distribution density ($Y$-axis) for samples in the MSC ($n = 115$) or non-MSC ($n = 510$) classes. Arrows indicate the scores that 99% of each class fall into. The overlap indicates the region of uncertainty, where a classification is given as 'unknown.' (F) A summary of the 16-gene MSC signature colour coded to the component (as described in 1B). Gene ID is given as HUGO symbol and ENSEMBL gene ID; C is component; P is probability of selection (indicating stability); R is correlation of gene to component (as per 1D); L is predicted subcellular location of Intracellular (I), Nucleus (N), Extracellular matrix (ECM), Secreted (S), Membrane (M) and U is unknown. See also Fig. S2 and Tables S2, S3.

to 510 definitively non-MSC samples from primary human tissues and cell lines, including cultured fibroblasts, haematopoietic cells and pluripotent stem cell lines (Tables S2 and S3).

To fully integrate and interrogate these data, we derived a novel cross-study analysis framework. Our approach, described in Fig. 2A, included a cross-platform normalisation step (*Lê Cao et al., 2014*), and a modified variable (gene) selection methodology. The first part of the protocol identified hundreds of potential MSC markers, which in combination greatly improved the classification accuracy of 97.7% (Table 1). This included many of the known MSC markers. Each gene was further evaluated for stability by subsampling the datasets to ensure that its inclusion was not reliant on one dominant source or platform. Stability is indicated by the probability of selection over 200 iterations in Fig. 2B, and was the step that excluded most of the commonly used MSC markers. For example, PDGFRB and VCAM1 were identified as potential component 1 genes but their inclusion was highly variable (0.76 and 0.59 probability of selection respectively).

We reasoned that if the majority of genes discriminating between MSC and non-MSC are describing a common biology and are highly correlated, then a subset of these genes could be identified that would represent the entire network. Therefore, we iteratively assessed how the inclusion of each gene contributed to the overall accuracy of the signature. This found the subset of variables that were most stable and least redundant at a statistical level, and that would represent the greater network of MSC-related measurements (Fig. 2C). Sixteen genes were identified, collectively forming a 'signature,' which provided a high degree of discrimination between MSC and non-MSC cell types, without any loss of accuracy in accurately identifying MSC (>95% correct MSC call or 4/125 misclassified MSC samples, Table 1) and with improved discrimination from fibroblasts and other non-MSC cell types (1.61% false positive, Table 1). We confirmed that this clustering was agnostic to technology platform or manufacturer (Fig. S2).

Cells derived from bone marrow were reliably grouped together with this method (Fig. 2D, Fig. S2E), and MSC from other tissue sources, including adipose tissue, skin, lung, placenta and cord blood shared this signature. Each gene in the signature made an additive

contribution across four vectors (components), such that the absolute expression of any one gene might differ from sample to sample but the combination of gene expression was highly predictive. High expression of component 1 genes was most likely to be a positive predictor of an MSC classification (Fig. 2 and Fig. S3A), as indicated by the correlation of expression of each gene with its component. Note that the components are linear vectors, and so a negative correlation (as for component 1 genes) simply indicates the contribution of the genes to clustering MSC on the positive or negative region of that component. The inclusion of components 2–4 provided higher discrimination for subsets of MSC and non-MSC, particularly differentiating MSC and fibroblasts derived from various tissues. These latter components included stress-related genes (heat shock proteins) and early indicators of lineage commitment (osteomodulin). Importantly, this multicomponent based approach, in contrast to a typical differential expression analysis, allowed for a common MSC phenotype that is also permissive of tissue-specific differences in the wider MSC gene network.

The implementation in www.stemformatics.org assessed the MSC score across 200 iterative predictions, where a sample must have a 95% pass rate to be classed as an MSC. The distribution of the training sample scores was used to determine high confidence scores (Fig. 2E). By using 200 subsamplings of the training set, 200 scores were recorded for each sample, which enabled us to derive an individual 95% Confidence Interval (CI). A sample was assigned to the MSC class if the lower bound of its 95% CI is strictly higher than 0.5169. Similarly, a non-MSC classification is given if the upper bound of the 95% CI was lower than 0.4337. Samples failing to meet these criteria were assigned to an 'unknown' category. Accordingly, the four misclassified MSC in the training set included one adult bone marrow MSC sample (predicted 1/200 times as MSC), and the remaining from two fetal studies, the first consisting of 10-week chorionic villi (predicted 29/200 times as MSC) and 12-week chorionic membrane preparation (2/200 MSC predictions), the second from a neonatal lung aspirate (0/200 positive MSC predictions).

## The MSC signature genes form a cohesive network implicated in healthy mesenchymal development and function

To assess possible functional relationships between MSC signature genes, we used a curated set of protein-protein interactions from the BioGrid database using the genes selected from component 1 that showed a high discriminating power between MSC and non-MSC. These formed a network of 36 interacting proteins (Fig. 3A). The higher expression of these genes in MSC samples is confirmed in Fig. 3B. If the statistical tool had identified a random set of genes, then the network would have little connectivity and there would be no relevant functional annotations. This was confirmed by random subsampling from the background datasets, which failed to form any PPI network. To assess whether the highly connected MSC network also shared any cohesive functional annotations, we examined mutation databases for evidence of human diseases associated with network members. A high proportion of the MSC network (30/43) are represented in Mendelian disorders of mesenchymal development by virtue of their mutation spectrum in facial or musculo-skeletal dysmorphologies in man, or evidence of mesodermal defects

in KO mouse models (Described in detail in Table S4). These included the paired-related homeobox-1 (*PRRX1*), a transcription factor important for early embryonic skeletal and facial development, and with a *de novo* mutation spectrum in the embryonic dysmorphology syndrome Agnathia-otocephaly (*Çelik et al., 2012*). Likewise, mutations in bone morphogenetic protein 14 (*BMP14/GDF5*) lead to developmental abnormalities in chondrogenesis and skeletal bone (*Degenkolbe et al., 2013*). Mutations in *DDR2* cause limb defects, including spondylo-epiphyseal-metaphyseal dysplasia (*Ali et al., 2010*) and mice over-expressing *DDR2* have increased body size and atypical body fat (*Kawai et al., 2014*). In humans, Polymorphisms in *ABI3BP* are associated with increased risk of osteochondropathy (*Zhang et al., 2014*), and mice lacking Abi3bp have profound defects in MSC differentiation to bone and fat (*Hodgkinson et al., 2013*).

We next examined functions that had been specifically validated in MSC biology, specifically, whether any members of the signature had been used to prospectively isolate MSC from tissue sources. *ITGA11* was a member of the core signature that has been used to prospectively enrich MSC from bone marrow with enhanced colony forming capacity (*Kaltz et al., 2010*), and independently shown to be enriched more than 3 fold at protein level in bone marrow MSC compared to dermal fibroblasts or perivascular cells (*Holley et al., 2015*). Although several of the known and commonly used MSC markers were indeed captured in the large initial set of potential classifiers, but rejected by our statistical method on the grounds of poor selection stability, these were 'rescued' in the protein interaction network. That is, the behavior of these markers was variable across laboratories and between microarray platforms, and often high expressed on non-MSC cell types. Nevertheless, the interaction network demonstrated some cohesive biology with these known markers. The most highly connected member of the extended network was VCAM1, which was identified in the large prospective marker set but with a low frequency of selection (0.6 on component 1), which eliminated it from the final classifier. VCAM1, together with STRO-1, has been used for the prospective isolation of human bone marrow MSC (*Gronthos, 2003*). VCAM1 is an adhesion molecule that is induced by inflammatory stimuli to regulate leukocyte adhesion to the endothelium (*Dansky et al., 2001*); however, in cardiac precursors its expression demarcates commitment to mesenchymal rather than endothelial lineages (*Skelton et al., 2014*).

Other members of our network that have been previously described in human or mouse MSC biology, and used to prospectively isolate cells or have been validated at the protein level include *PDGFRβ* (*Koide et al., 2007*), *SPINT2* (*Roversi et al., 2014*), *CCDC80* (*Charbord et al., 2015*), *FAP* (*Bae et al., 2008*), *BGN* (*Holley et al., 2015*),and *TM4SF1* (*Bae et al., 2011*). SPINT2 is a serine protease inhibitor whose activity is required in bone-marrow MSC, and its loss alters hematopoietic stem cell function in myelo-dysplastic disorders (*Roversi et al., 2014*). In mice, CCDC80 is also necessary for reconstitution of bone marrow and support of haematopoiesis (*Charbord et al., 2015*).

The network included a high proportion of extracellular proteins (54%) with demonstrated roles in the modification of extracellular matrix proteins including proteoglycans, as well as regulators of growth factor and cytokine signalling. This included the cell migration inducing protein (KIAA1199/CEMIP), which is secreted

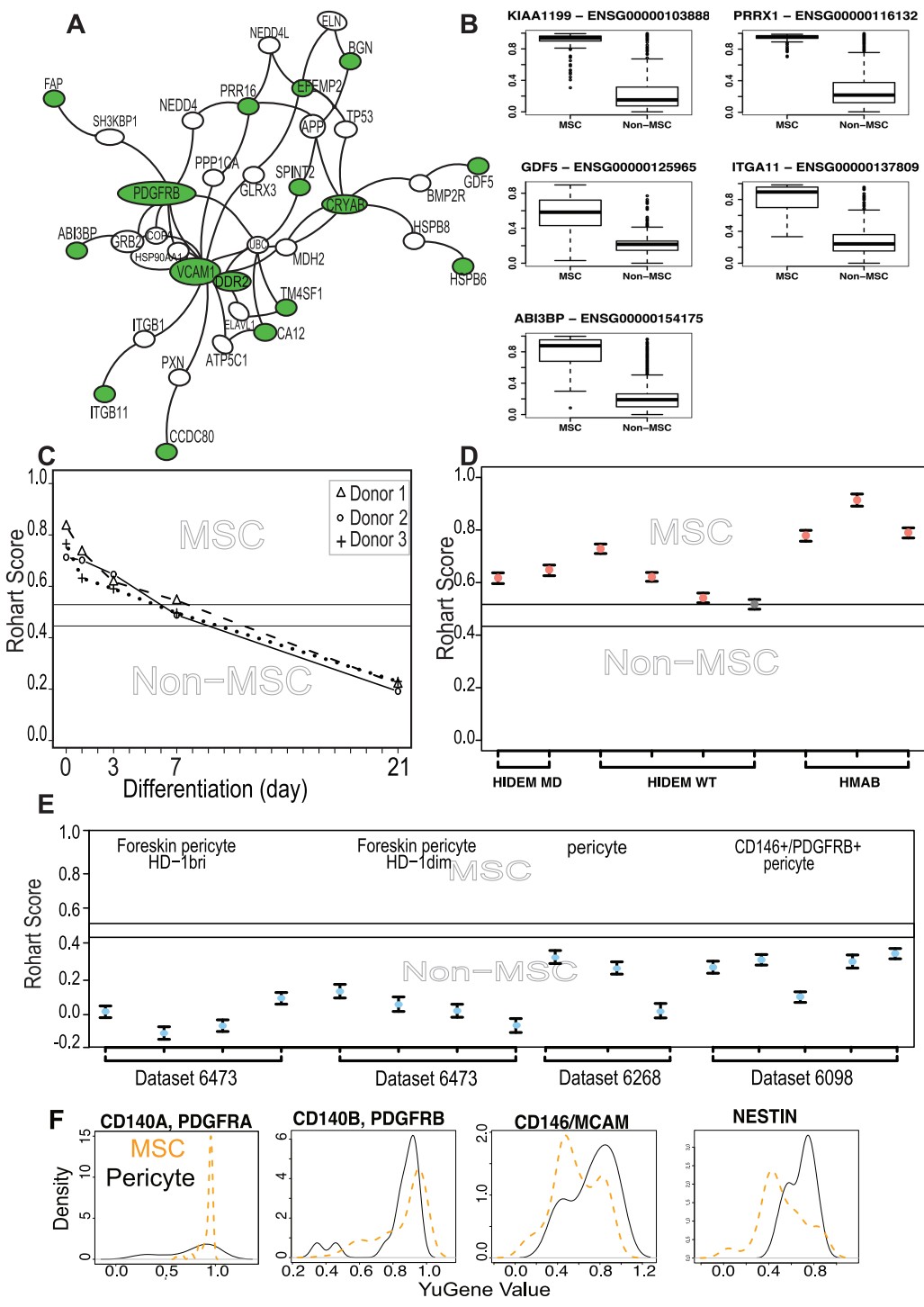

**Figure 3** **The MSC signature forms part of a network of extracellular proteins and discriminates between differentiating or related adult stem cell types.** (A) An extended protein-protein network diagram of the Rohart MSC signature genes demonstrating a role for VCAM1 and PDGFRB as part of a functionally interconnected set of glycoproteins, integrins, growth factors and extracellular matrix proteins. Green nodes are seed network members from component 

**Figure 3 (…continued)**
1 genes, white nodes are inferred network members, and edges are protein-protein interactions. (B) Box and Whisker plots showing average expression of the genes making up the MSC signature component 1 genes in MSC ($n = 115$) and non-MSC ($n = 510$). Note, PRRX1, GDF5, ITGA11 and ABI3BP also form seeds in the network. KIAA1199 lacks PPI data and is not annotated in the network. (C) Classification of bone marrow MSC over a time course of differentiation to cartilage; $y$-axis gives the Rohart score, $x$-axis orders the samples from each experimental series. Three differentiation series from three donors are shown. The uncertainty region stands between the MSC and non-MSC prediction regions. (D) Classification of perivascular-derived stem cells from skeletal muscle mesangioblasts (HMAB), or iPSC-derived mesangioblasts (HIDEM) from donors with muscular dystrophy (MD) or healthy donors (WT). Error bars around each prediction score represent the CI boundaries. A sample is classified as 'unsure' (indicated in grey) if its prediction score or its CI overlapped the uncertainty region. (E) Classification of pericytes derived from three distinct datasets: from Left–Right neonatal foreskin (Antigen HD-1 dim or bright); placental pericytes; perivascular endometrial stem cells (CD146+/PDGRFB+). Stemformatics dataset identifiers provided for each experimental series. Error bars around each prediction score represent the CI boundaries. (F) Distribution of expression of common MSC/Pericyte markers. $X$-axis is Gene expression ranked by the YuGene cumulative proportion, $Y$-axis is the density distribution of MSC (orange plot, $n = 115$) or pericytes (black plot, $n = 16$). See also Fig. S3 and Tables S4, S5 and S6.

in its mature form. It regulates Wnt and TGF$\beta$3 signalling by depolarising hyaluronan, and may alter trafficking of cytokines and growth factors to the extracellular milieu (Yoshida et al., 2014). DDR2 is a receptor tyrosine kinase that interacts directly with collagens. It stabilises the transcription factor SNAIL, and has been implicated in epithelial-mesenchyme transitions in epithelial cancers (Zhang et al., 2013). CCDC80 binds syndecan-heparin sulphate containing proteoglycans, has been shown to inhibit WNT/beta-catenin signalling and has a regulatory role in adipogenesis (Tremblay et al., 2009; Walczak et al., 2014). SRPX2 is a secreted chondroitin sulfate proteoglycan involved in endothelial cell migration, tissue remodelling and vascular sprouting (Royer-Zemmour et al., 2008). The chaperonins HSPB5/CRYAB and HSPB6 stabilise protein complexes, and may assist in delivery of growth factor complexes where these are present in high concentrations. In transplantation paradigms it is likely that the therapeutic benefit derived from MSC is via local immunomodulatory, anti-inflammatory, and/or trophic effects during the acute phase of cell therapy. The network of genes identified here as enriched in MSC suggests an over-arching role for these cells in modifying the extracellular environment, functions important in development as well as in homeostatic regulation of adult tissues.

## MSC differentiation, dedifferentiation and the MSC signature

The majority of public microarray datasets available to us had limited phenotypic data available, so these were not used to derive our MSC signature. Nevertheless we annotated each of these samples as *presumptive* MSC (213 samples) or *presumptive* non-MSC (499 samples) based on their origin and use in the source publication (Table S5). Where MSC were profiled during *in vitro* lineage differentiation, we assigned the samples taken at intermediate time points to an 'unknown' category (579 samples) prior to testing these with the signature. Implementation of the Rohart Test in the www.stemformatics.org resource allowed us to evaluate a wide range of different experimental paradigms. Despite the lack of phenotypic information associated with these datasets, the agreement between

publication status and our classification was high. Five percent of the *presumptive* non-MSC (27/499) were misclassified by the signature as MSC, and around half of these (>13) were neonatal or fetal dermal fibroblasts (Table S5). Others have reported MSC fractions derived from dermal tissues (reviewed in *Vaculik et al., 2012*) and certainly fibroblasts from other sources were not classified as MSC. Furthermore, the signature could discriminate between MSC and differentiating cultures. Figure 3C demonstrates loss of the MSC score during chondrogenic differentiation with the addition of TGF$\beta$ (Dataset 6119; *Mrugala et al., 2009*) and this pattern was recapitulated for cells differentiating to mineralising bone (Data not shown, but the reader is referred to the Stemformatics resource, see: https://www.stemformatics.org/workbench/rohart_msc_graph?ds_id=6206#) or to adipose-like cells (https://www.stemformatics.org/workbench/rohart_msc_graph?ds_id=6208#) or when undergoing reprogramming of an adipose-tissue derived iPSC (https://www.stemformatics.org/workbench/rohart_msc_graph?ds_id=5018).

## Comparison of MSC and adult stem/progenitor cell types

The limbal cell niche hosts both limbal epithelial and stromal progenitors (*Lim et al., 2012*), and the stromal progenitors were also classified as MSC by our tool (Dataset 6450). Some MSC subsets are likely to be derived from a perivascular progenitor. In our hands, primary skeletal-muscle mesoangioblasts thought to be a subset of perivascular cells in skeletal and smooth muscle (Dataset 6265: *Tedesco et al., 2012*), defined as alkaline-phosphatase$^+$ CD146$^+$ CD31/Epcam$^-$ CD56/Ncam$^-$ with demonstrated skeletal muscle differentiation, were classified as MSC (Fig. 3D). In contrast, the majority of cells derived from a perivascular location (and confirmed as such with tissues imaged in the source publication) were not classified as MSC (Fig. 3E). On examining putative markers of perivascular progenitors in these samples, we could demonstrate that the majority of perivascular progenitors expressed higher levels of Nestin than the majority of MSC (Fig. 3F). MCAM+ and MCAM− cells were apparent in both MSC and pericyte groups, although a higher proportion of perivascular progenitor expressed MCAM RNA. In contrast, PDGFRA was highly expressed in MSC but not informative in perivascular cells, and PDGFRB was highly expressed in both populations. Others have shown that high expression of PDGFRA is associated with highly proliferative MSC colonies, suggesting that its expression is associated with expansion in culture (*Samsonraj et al., 2015*). These data are consistent with a classification hierarchy determined by mouse and human lineage studies, where multipotent adult cells are quiescent in a perivascular location (*Crisan et al., 2008*; *Acar et al., 2015*). Thus perivascular progenitor cells with MSC differentiation capacity are defined as Rohart test negative, Nestin positive in our test, and as such are distinct from a Rohart test positive MSC. Cells differentiating to osteoblast, chondrocyte, adipocyte or fibroblast exit the MSC state and rapidly become negative for the Rohart MSC score. Given that a proportion of Rohart test positive MSC express MCAM or Nestin, the classification tool may detect a phenotypic spectrum that spans the intermediates across the perivascular-MSC-fibroblast hierarchy.

## Tissue clustering of MSC is confounded by sex and MHC-1 haplotype

The capacity to group MSC-like cells is consistent with the general assumption that MSC from different tissue share some common molecular properties. Many of the individual studies in this reanalysis describe tissue-specific differences in MSC populations. We were not able to recapitulate any of these specific differences on the integrated dataset. Nevertheless, MSC from different tissues did form subclusters (Figs. S2 and S3), and the majority of bone marrow MSC clustered together (Fig. S2E). We therefore examined more broadly the genes that were significantly different between bone marrow MSC and other cell types at the whole transcriptome level. This analysis confirmed the observed clustering of bone marrow derived MSC, distinguished by differential expression of 425 genes (adjusted $P < 0.01$, Table S6). The genes that were most differentially expressed between the different MSC sources in our combined analysis were MHC class I genes, and these accounted for >40% of the top 100 differentially expressed genes in the bone-marrow comparisons (Table S6). The HLA isotypes were generally, but not exclusively, expressed at lower levels in bone marrow MSC (Hierarchical Cluster, Fig. S3). Estrogen and progesterone receptors, and a network of associated target genes were also significantly different between tissue sources (Table S6), and this may reflect a bias in the sex of the donors from which tissue was sampled; although the sex of the donors was not available for a majority of samples, some tissues (such as decidual sources) will be entirely female in origin. Further molecular sub-classifications of MSC will therefore require much larger studies that address specific clinical or differentiation properties of the cells, and must also consider ascertainment biases that may introduce confounding variables such as HLA subtypes or sex.

## DISCUSSION

Modern molecular classification tools are needed for the characterisation of MSC *ex vivo* and *in vivo*. Antibody based methods currently rely on a subset of cell surface proteins that are widely acknowledged to lack specificity, and the reliability of these assays is dependant on operator expertise. Our study set out to provide an alternate test that had better discrimination power than current assays, was robust and easy to generate. In doing so we developed a specific gene signature that is shared by a wide-variety of MSC. The "Rohart MSC test" is an *in silico* tool that has been implemented as a simple online test that will be useful in standardisation or improvement of current bulk isolation methods. This classification tool is available in the www.stemformatics.org platform, together with all the primary data used in derivation of the signature. Details on submitting proprietary data to the Rohart test are available on the www.stemformatics.org site.

All together we curated more than 120 MSC-related gene expression datasets in the www.stemformatics.org resource (*Wells et al., 2012*); the datasets can be queried here using key word, dataset ID or author, together with an implementation of the Rohart MSC test.

Our approach highlights the potential robustness of biological signatures when combining data from many different sources, where experimental variables such as platform or batch can be reduced (Fig. S2). The methods we used for derivation of a common MSC classifier could be applied to the meta-analysis of any cell subset or phenotype where sufficient samples can be drawn from public expression databases.

The Rohart test provides a snap shot of the current state of play in MSC biology. As an *in silico* test it reflects all of the ambiguities existing in current nomenclature and culture practise. We anticipate that a computational classifier will evolve as the field of MSC biology evolves, and as isolation methods improve. Indeed, the question of what is an MSC, and whether these are a distinct stem cell population recruited from the bone marrow, as suggested by mouse studies of fetomaternal microchimerism (*Seppanen et al., 2013*) or from perivasculature, as suggested by immunotagging of MSC-like cells from perivascular regions in human tissues (*Crisan et al., 2008*), or are resident progenitor populations specific to each organ cannot be resolved in the current study. The signature itself is dependent on the quality of the MSC used in the training set. As rare adult stem/progenitor cell types were under-represented in the current test or training datasets, we anticipate that functional classification of MSC subtypes will improve as newer sampling methods provide the means to identify and replicate these cells. To highlight this point, the signature distinguishes perivascular progenitors from MSC, however resolving a perivascular progenitor signature would require substantially more data on this population than is currently available in the public domain. We expect that further refinements in the isolation or culture of purer MSC or more precisely defined functional subsets will also result in future evolutions of this *in silico* signature.

In summary, we set out to systematically review the current state of play in MSC biology using a meta-analysis of transcriptome studies, and in doing so were able robustly to identify a general MSC phenotype that could distinguish MSC from other cell types. The resulting signature could also identify points of transition as MSC underwent differentiation or reprogramming studies. Furthermore, we demonstrated that, at least at a gene expression level, our *de novo* derived signature outperformed the classification accuracy of the combined set of traditional MSC cell surface markers. While a signature approach such as ours is not able to resolve the ontogeny or *in vivo* function of MSC, it does provide a tool for better benchmarking and comparison of the cells grown *ex vivo*, and will assist with comparison of cells derived for clinical purposes. The methods that we describe here, and the resulting molecular classifier, represent an important step towards addressing the more intractable questions of MSC identity, ontogenic relationships and function.

### Funding
This work was funded by an Australian Research Council Grant SR1101002 to Stem Cells Australia (CAW), ARC discovery project DP130100777 to CAW and KALC, JEM Research Foundation philanthropic funding to CAW, and an NHMRC project grant APP1023368 to NMF and KK. CAW is funded by a QLD Government Smart Futures Fellowship. KK was supported by NHMRC career development fellowship 1023371. JP was supported by the National Heart Foundation Australia. KALC is funded in part by the Australian Cancer Research Foundation (ACRF) for the Diamantina Individualised Oncology Care Centre

at the University of Queensland Diamantina Institute. The funders had no role in study design, data collection and analysis, decision to publish, or preparation of the manuscript.

## Grant Disclosures

The following grant information was disclosed by the authors:

Australian Research Council: SR1101002.

ARC discovery project: DP130100777.

JEM Research Foundation.

NHMRC: APP1023368.

NHMRC career development fellowship: 1023371.

National Heart Foundation Australia.

Australian Cancer Research Foundation (ACRF).

## Competing Interests

The authors declare there are no competing interests.

## Author Contributions

- Florian Rohart conceived and designed the experiments, analyzed the data, wrote the paper, prepared figures and/or tables, reviewed drafts of the paper.
- Elizabeth A. Mason performed the experiments, prepared figures and/or tables, reviewed drafts of the paper, determined the criteria for MSC annotation, implemented systematic review of MSC literature.
- Nicholas Matigian performed the experiments, reviewed annotations and confirmed review of literature.
- Rowland Mosbergen, Othmar Korn and Tyrone Chen analyzed the data, reviewed drafts of the paper, developed the software resource to host the MSC test.
- Suzanne Butcher performed the experiments, reviewed drafts of the paper, annotated datasets for MSC criteria.
- Jatin Patel, Kerry Atkinson and Kiarash Khosrotehrani contributed reagents/materials/-analysis tools, reviewed drafts of the paper.
- Nicholas M. Fisk contributed reagents/materials/analysis tools, wrote the paper, reviewed drafts of the paper.
- Kim-Anh Lê Cao conceived and designed the experiments, analyzed the data, wrote the paper, reviewed drafts of the paper.
- Christine A. Wells conceived and designed the experiments, analyzed the data, contributed reagents/materials/analysis tools, wrote the paper, prepared figures and/or tables, reviewed drafts of the paper.

## Data Availability

In total, 64 in-house derived samples contributed to this meta-analysis.

Forty-five microarray profiles of gestational MSC sources were deposited in ArrayExpress as E-TABM-1224 (http://www.ebi.ac.uk/arrayexpress/experiments/E-TABM-1224/) and the processed data used in this meta-analysis is also available at http://www.stemformatics.org/datasets/search?ds_id=6063.

A further 12 microarray samples from diverse MSC sources was deposited at Array Express E-TABM-880 (http://www.ebi.ac.uk/arrayexpress/experiments/E-TABM-880/) and the processed data used in this meta-analysis is also available at http://www.stemformatics.org/datasets/search?ds_id=6064.

A further seven endothelial progenitor cell datasets were deposited in ArrayExpress (http://www.ebi.ac.uk/arrayexpress/experiments/E-MTAB-3277/) and the processed data used in this meta-analysis is also available at http://www.stemformatics.org/datasets/search?ds_id=6306

The code described in this manuscript is available in the CRAN repository under the bootsPLS package (https://cran.r-project.org/web/packages/bootsPLS/index.html).

## Supplemental Information

Supplemental information for this article can be found online at http://dx.doi.org/10.7717/peerj.1845#supplemental-information.

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
