# Peer review of "A molecular classification of human mesenchymal stromal cells"

_PeerJ, doi:10.7717/peerj.1845_

## Round 0.1 · original submission · Minor Revisions

Dear Professor Wells,

I have now received three reviews of your manuscript “A molecular classification of human mesenchymal stromal cells”. Two reviewers have highlighted a number of data omissions and sections in the manuscript that require more explanation of the reasoning behind choosing the criteria used in the study. Of particular note are discrepancies in numbers in tables. Having dealt with many high-throughput datasets myself I can appreciate that it is a lot more difficult to get these types of details sorted for high-throughput data than in standard studies; however, it will be important to fix the list of issues picked up on by reviewers 2 and 3, all of which seem reasonable, before the manuscript could be considered acceptable for publication. While there are a lot of issues to address, they do not require additional experimentation and so strike me as very easily and quickly doable, perhaps even within a week. Thus I would welcome a revised manuscript within the next 40 days.

Best, Eric

Reviewer 1 ·

Basic reporting

This manuscript by Rohart et al presents an exhaustive analysis of the transcriptome of a large number of MSCs, cells presumed to be MSCs and other cell types such as fibroblasts in order to provide a molecular classification of human MSCs based on a set of stringently defined markers. The result is a novel bioinformatics tool, a classifier which robustly distinguishes MSCs from non-MSC cell populations. Timing of this study is on the spot because a large number of transcriptome datasets are now available with no information on their consistency between platforms of labs. The work by Rohart et al now takes care of this and provides an extremely useful evaluation of the markers used for MSC classification. The tool developed by Rohart et al. further leads to the identification of gene sets that can reliable define an “MSC type”
The study is very well designed and criteria selection well rationalized. The training sample sets are large, allowing for robust statistics. It is clear that the authors have put tremendous effort into this work. In addition to their informative analyses, the authors also provide interesting conclusions, such as e.g. the importance of considering both mRNA and protein expression data, when available. I find the study well conducted, well documented, well written and extremely useful to the field of not only stem cells but also gene regulation in general. I cannot find anything negative to say about this study.
One obvious question emanating from this work is whether the Rohart classifier has any predictive function; this would be reasonable to assume given the robustness of the data presented. This could be the aim of future work.
In summary, the authors should be commended on this terrific work which in my opinion should be published immediately.

Experimental design

Well thought of. State of art.

Validity of the findings

Strong.

Additional comments

All comments are included in the first statements.

Reviewer 2 ·

Basic reporting

1) Data missing (e.g. Table S3)
2) Not enough information was given about figures and tables
3) Figure/table used has nothing to do with the text (e.g. line 278 and Sup Table S4)
4) Inconsistency of the style of referencing (e.g. in Introduction v.s. the rest of text)
5) typo throughout the manuscript

Experimental design

1) The selection of microarray datasets was not clear
2) The criteria that the authors decided to use seemed to be arbitrary e.g. cell surface markers CD105+, CD73+, CD45- (e.g. why these 3 and not others?) And also why differentiation to at least 2 of the three MSC- definity lineages? (and not all three?)

Validity of the findings

1) Insufficient data/explanation was provided to support the conclusion in the result section

Reviewer 3 ·

Basic reporting

The manuscript uses clear and professional language throughout the text. The introduction fairly well presents the problem with MSC marker heterogeneity, and although one may argue that lines 182 to 187 present the main objective of the work and belong in the introduction, I do not subscribe to that argument.
Making the statistical model and code available, lines 123-127, is a strong positive for this work.
All tables and figures are relevant. Suggestions for improving Figure 3 are offered (see below).

Experimental design

The research question is well defined on lines 182-187 of the submission. The work was performed to high standards.
Were any of the criteria by which the gene expression datasets evaluated in the majority or much more commonly occurring than others? If so, please provide these in line 72.
Please provide operating parameters and version numbers of software used in lines 99 to 102.
Please give the date (or version number) on which the IMEx PPI data were accessed. How many different randomized sets of proteins were used (to attempt) to build comparative PPI networks in lines 134-136?

Validity of the findings

Network construction. The text on line 268 states that the network has 43 interacting proteins, but only 36 are presented in Fig 3A. Furthermore, some of the genes for which expression data are presented in Fig 3B are not presented in the network, but offered as evidence that those genes exhibit higher expression in MSC samples. First, please present the entire network of 43 proteins, or revise the text to state that the network is of a different size. Second, make certain that the presented gene expression data represent genes found in the network diagram.
The top differentially expressed genes between MSC from bone marrow compared to other sites, Table S6, was not present in the files made available for review.
The conclusion is stated well and relates to the objective of the work. Areas where the presented work can be improved (sex and HLA aspects, eg) or where future work may be necessary are also presented.

Additional comments

Impact is a noun and should not be used as a verb, especially in formal and scientific writing
Data should be used in the plural; see lines 97 and 202 for example, and elsewhere
Sometimes the authors use datasets and other times data sets. Make this consistent.
Line 32: a single functional standard is given, thus criterion
Lines 43-44: “line-line variation” borders on jargon. Do the authors intend to offer variation across cell lines as a prime contributor?
Lines 78-81: This is a run-on sentence. Either use a conjunction or make into two sentences.
Line 258: 10-week?
Lines 357, 372: missing )
Line 398: tissue or tissues?
Use sex in lines 414-415, 420 in place of gender. Sex is biological, which seems to be the important characteristisation here, and gender is more toward one’s image of oneself.

---

## Round 0.2 · accepted · Accept

I have read through the manuscript and the track changes myself and see that all issues raised by reviewers 1 and 3 and most by reviewer 2 have been satisfied. For reviewer 2, I think some comments may have reflected missing information that was embedded in the supplemental files. While I think the goal of this reviewer may have been for these to be stated in the main text, I am ambivalent on this issue because having it in the text makes it easier on an interested reader in the field who wants these details but makes the text harder to follow for a reader interested in the principles and final outcome. Thus I think it is adequate to keep this information in the supplemental files. I would note that in going in detail over the manuscript I noticed that sometimes supplementary files are referred to as "supplementary figure 1" and sometimes as "supplementary figure S1" and the same for tables. This may have caused some of the reviewer confusion earlier regarding the files and should be corrected, but as it is quite minor I think it can be fixed at the proof stage.